# Convergent extreme reductive evolution in ancient planthopper symbioses

Anna Michalik ®[1] ✉, Diego C. Franco[1,2], Junchen Deng[2,3],
Monika Prus-Frankowska[2], Adam Stroiński[4] & Piotr Łukasik ®[2] ✉

Strictly heritable endosymbiotic bacteria that provide limiting nutrients to sap-sucking hemipteran insects are known for their highly reduced genomes conserved in organization and function. Here, we show how in ancestral endosymbionts of planthoppers, *Sulcia* and *Vidania*, which have been gradually losing genes during ~263 my of co-diversification with hosts, co-infections by additional microbes and host ecological switches coincided with more dramatic genomic changes. At its extremes, this has resulted in the smallest non-organellar bacterial genomes known, at barely 50-52 kb. Such minuscule *Vidania* genomes evolved convergently in two planthopper superfamilies, and are strikingly similar in gene contents, including the ability to produce a single amino acid (phenylalanine) for the host. Losing many additional cell-function genes places them very close to organelles of symbiotic origin in the level of host dependence, further blurring the bacteria-organelle boundary.

The smallest artificial, carefully designed bacterial genome capable of axenic growth in a rich medium is only 531 kilobases [kb], an order of magnitude less than that of the familiar *Escherichia coli*[1]. However, for some two decades, we have known that the genomes of obligatorily intracellular insect-symbiotic bacteria that rely on host support mechanisms can be much smaller[2]. Specifically, the genomes of ancient, strictly heritable bacterial symbionts in diverse lineages of sap-sucking hemipteran insects have convergently reduced to only encode genes involved in essential nutrient biosynthesis and some key cellular functions, and sometimes barely exceed 100 kb[3–9]. Nevertheless, after the presumed early period of extensive genomic degeneration, several symbiont lineages have reached the stage where the remaining genes are indispensable, and their further loss slows to nearly a halt[9]. This is demonstrated by the conservation of gene sets and order among divergent strains of *Buchnera* or *Tremblaya*, which have co-diversified with their respective hemipteran hosts for over 200 my[10,11]. However, stability has its limits, and surveys encompassing a greater range of host lineages have revealed lost genes and occasional rearrangements[11,12] or more dramatic processes such as the extreme and rapid degeneration of the

symbiont *Hodgkinia* in some cicadas[5,13]. These unusual genome evolutionary processes in hemipteran symbionts provide a unique natural framework for studying the boundaries of cellular life and the long-term consequences of obligate intracellular existence. Yet, although the overall patterns of gene loss and functional streamlining are established[6,9], it remains unclear how far this process can proceed, what factors govern it, and to what extent different symbiont lineages follow parallel evolutionary trajectories. Addressing these questions requires systems that encompass deep evolutionary timescales, ancient symbiotic associations, and multiple independent changes to microbial communities and to host ecology.

Here, we investigate the evolutionary dynamics of symbiont genomes in planthoppers (hemipteran infraorder Fulgoromorpha), a phylogenetically diverse, ecologically significant, and economically important clade of sap-feeding insects that originated ~263 million years ago. Planthoppers host a broad array of heritable nutritional endosymbionts, including ancient associations with two strictly vertically transmitted bacterial lineages with highly reduced genomes—the Bacteroidetes symbiont *Candidatus* Sulcia muelleri (hereafter *Sulcia*) and the Betaproteobacterium *Candidatus* Vidania

[1]Department of Developmental Biology and Invertebrate Morphology, Institute of Zoology and Biomedical Research, Faculty of Biology, Jagiellonian University, Kraków, Poland. [2]Institute of Environmental Sciences, Faculty of Biology, Jagiellonian University, Kraków, Poland. [3]Doctoral School of Exact and Natural Sciences, Jagiellonian University, Kraków, Poland. [4]Museum and Institute of Zoology, Polish Academy of Sciences, Warsaw, Poland. ✉e-mail: a.michalik@uj.edu.pl; p.lukasik@uj.edu.pl

fulgoroidea (*Vidania*)[12,14,15]. In many planthopper clades, these ancestral partners have been complemented or replaced by other bacteria or fungi. This diversity provides an unparalleled opportunity to examine how extreme reductive evolution proceeds across related yet independently evolving symbioses and to test general principles governing the structure, stability, and functional limits of nutritional endosymbiont genomes. Here, using metagenomic data from 149 planthopper species representing 19 families[14], we identify the processes, limits, and drivers of genomic evolution of their ancient symbionts.

In this work, we show that the ancient bacterial symbionts *Sulcia* and *Vidania* of planthoppers are generally highly stable in their genomic organization and contents, yet some lineages exhibit unprecedented genome reduction, yielding the smallest bacterial genomes known to date, at only ~50–52 kilobases. We demonstrate that these ultra-small genomes, which have evolved convergently in distinct planthopper lineages, can encode nearly identical sets of essential genes. Comparative analyses indicate that host ecological shifts and the replacement or supplementation of ancient symbionts by additional microbial partners are the primary drivers of extreme genome erosion. Together, our results delineate the lower boundaries of bacterial genomic complexity and illuminate the evolutionary forces that shape the transitions of ancient, specialized symbioses to even more reduced and host-integrated forms.

## Results

### *Sulcia* and *Vidania* have co-diversified with planthoppers for ~263 my

*Sulcia* and *Vidania* infections are broadly distributed across planthopper taxonomic diversity (Fig. 1). Based on rRNA sequences reconstructed from metagenomes, we have detected *Vidania*, usually accompanied by *Sulcia*, in 87 species of 149 sequenced, representing 15 of 19 surveyed families, on both sides of the deepest split in the planthopper phylogeny (Fig. 1, and Supplementary Data 1). Indeed, *Sulcia* infection is known to have pre-dated the divergence of planthoppers from their sister clade, Cicadomorpha, which includes cicadas, spittlebugs, leafhoppers, and treehoppers[16]. In turn, the initial infection with *Vidania* seems to have occurred after these clades separated[12]. *Sulcia* and *Vidania* have co-diversified strictly with their planthopper hosts (Supplementary Fig. 1), as expected given their transovarial transmission[17]. Where both *Sulcia* and *Vidania* are missing, we always observe Hypocreales fungi—members of an insect-pathogenic clade known to have repeatedly replaced ancient bacterial nutritional endosymbionts of diverse Auchenorrhyncha, including planthoppers[18–20].

Further, we often find Gammaproteobacteria (*Sodalis*, *Arsenophonus*, *Symbiopectobacterium*) and Alphaproteobacteria (*Wolbachia*, *Tisiphia*, *Rickettsia*, Bartonellaceae, Acetobacteraceae), acquired independently by different host lineages[17,19,21,22]. These diverse symbionts are known to provide vitamins and other complementary

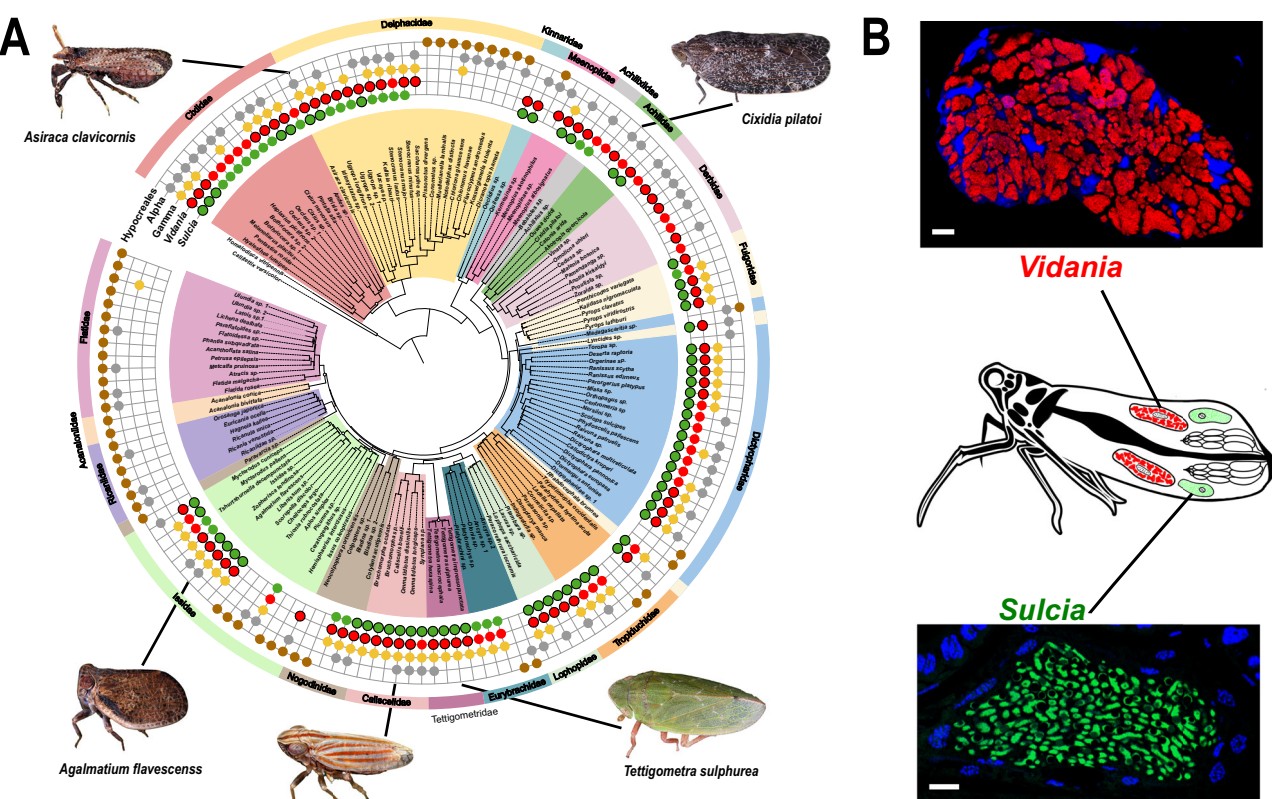

**Fig. 1 | The distribution of the ancient symbionts *Sulcia* and *Vidania* across planthopper phylogeny, and the organization of symbionts within host tissues. A** Two ancient symbionts, *Sulcia* and *Vidania*, are broadly distributed across the taxonomic diversity of planthoppers. The planthopper phylogeny is based on 1164 nuclear and 13 mitochondrial genes, after Deng et al. 2025[14]. Red and green circles represent strains of *Vidania* and *Sulcia*, respectively, and bold outlines indicate genomes that were fully assembled and used for subsequent analyses. Yellow, grey, and brown circles indicate the presence of Gammaproteobacteria, Alphaproteobacteria, and Hypocreales fungi, respectively. **B** *Sulcia* (green) and *Vidania* (red) reside within the cytoplasm of specialized host cells (bacteriocytes) that form dedicated organs (bacteriomes) within the host body cavity. Here, *Akotropis quercicola* (Achilidae) exemplifies the conserved symbiont morphology and bacteriome organization across taxonomically diverse planthoppers. The images were selected as representative from multiple images obtained for different planthopper species hosting *Sulcia* and *Vidania* symbionts, with replicate specimens for each species. Confocal microscopy, scale bar – 10 μm, blue represents DAPI staining. Photo credit: G. Kunz (insects).

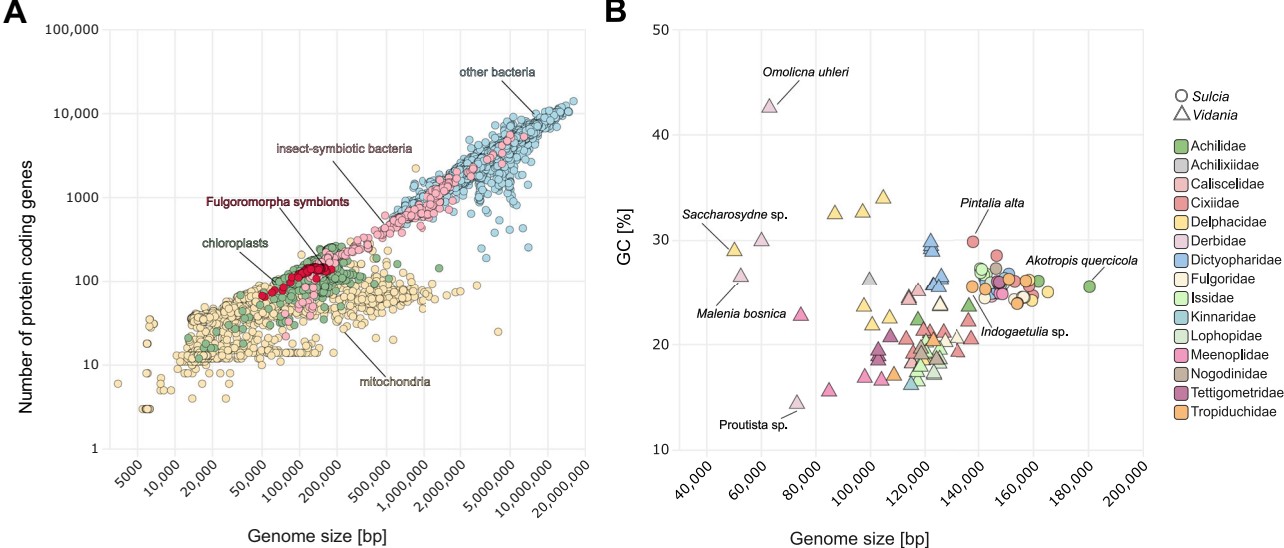

**Fig. 2 | A comparison of genomic characteristics among ancient planthopper symbionts *Sulcia* and *Vidania*, relative to other bacteria and organelles. A** In terms of size and gene content, Fulgoromorpha symbiont genomes (red) fall among the very smallest known for insect-symbiotic bacteria (pink), which themselves are among the smallest bacterial genomes overall (other, non−insect-symbiotic bacteria shown in blue). Cellular organelles−mitochondria (orange) and chloroplasts (green)−are shown for reference. The data underlying the scatterplot represent 66949 complete genomes downloaded from NCBI in April 2024, listed in Source Data. **B** In terms of size and GC contents, the newly reconstructed genomes of planthopper-associated *Sulcia* tend to be relatively consistent, whereas *Vidania* genomes represent a much broader range on both scales. Source data are provided as a Source Data file.

nutrients[17,19,21–23] but also influence host reproduction, defense, or other traits[24,25]. Together, our data show the relative stability of ancient bacterial associations while highlighting the evolutionary dynamics of more recently acquired microbes.

### *Sulcia* and *Vidania* have tiny genomes

Of 131 complete genomes of planthopper symbionts (63 *Sulcia*, 67 *Vidania*) that we used for content comparisons, most were circular. The two exceptions included the genome of *Sulcia* from *Trypetimorpha occidentalis*, previously confirmed to comprise two distinct circularly mapping contigs[21], and *Sulcia* from *Brixia* sp., with the same organization. Analysis of read mapping and the remaining contigs in the assemblies confirmed the completeness of genomes, as did synteny comparisons. Specifically, all but one of *Vidania* genomes, including the two smallest, were syntenic relative to the ancestral state[12] (Supplementary Fig. 2). In *Sulcia*, one or more rearrangements relative to the ancestral state were seen in 13/63 (20.6%) genomes representing six families (Supplementary Fig. 3). We note that some of the *Sulcia* and *Vidania* genomes that we could not fully assemble using short-read data, with insufficient DNA amount and quality for long-read sequencing, also appear to have undergone structural changes.

The reconstructed *Sulcia* genomes ranged in size from 137,729 bp to 180,379 bp, and *Vidania* ranged from 50,141 bp to 136,554 bp (Supplementary Data 2). Several newly assembled *Vidania* genomes are much smaller than the tiniest bacterial genomes characterized so far, including *Nasuia* from leafhoppers (≥107.8 kb), previously characterized *Vidania* (≥108.6 kb), or *Tremblaya* from mealybugs (≥139.0 kb)[7,10,21,26]. Correspondingly, many *Vidania* strains encode fewer protein-coding genes than almost any other symbiont - as few as 62 in one case (Fig. 2A). Smaller gene sets have so far only been reported from *Hodgkinia* that form unique complexes of cytologically and genetically distinct but complementary lineages that apparently exchange gene products[5,13,27], and from organelles: mitochondria and some of the most reduced chloroplasts[28,29]. *Vidania* genomes were much more variable in size, gene set, and GC content than those of *Sulcia* (Fig. 2B).

## Variable rate of symbiont gene loss during co-diversification with planthoppers

The contents comparison among these unusual and, in the case of *Vidania*, extremely rapidly evolving genomes (Supplementary Note 1) enabled the reconstruction of their ancestral gene sets. In the last common ancestor of extant planthoppers, *Sulcia* must have encoded ≥164 total genes with identified functions. This includes 9 genes involved in the biosynthesis of three essential amino acids (Leu, Ile, Val) plus one gene for Phe biosynthesis, 132 genes involved in genetic information processing, and 18 involved in metabolism. The ancestral *Vidania* must have encoded ≥169 total genes, including 43 genes for the biosynthesis of seven amino acids (Met, Arg, Thr, Trp, Phe, Lys, His), 113 genes for genetic information processing, and 13 metabolism-related (Supplementary Figs. 4 and 5). Additionally, across all *Sulcia* and *Vidania* genomes, we identified 40 and 86 open reading frames with no significant similarity to any records within the Uniprot database. While some of them were conserved in groups of species, suggesting functionality (Supplementary Data 3 and 4, and Supplementary Note 1), we did not include them in subsequent analyses.

We found no cases of apparent gene gain by any lineage. However, gene losses were common during the evolution of both *Sulcia* and *Vidania*, with a notably higher loss rate in the latter (Fig. 3). In both symbionts, the patterns suggest a gradual loss of genetic information processing and metabolism genes during host-symbiont co-diversification. Some genomes lost <5% of the reconstructed ancestral gene set, but we observe several instances of much more substantial losses involving also nutrient biosynthesis pathways (Fig. 3A, B, and Supplementary Figs. 4 and 5). For *Vidania*, this includes 49 gene losses on the branch leading to *Saccharosydne* sp. (Delphacidae), 40 losses in the lineage leading to the sole characterized Achilixiidae, *Bebaiotes* sp., 40 in the common ancestor of Derbidae, 32 in the ancestor of *Meenoplus skotinophilus* (Meenoplidae), and 25 in the ancestor of *Cixidia pilatoi* (Achilidae). Strikingly, four of the five lineages where *Vidania* lost most genes (all listed above except *Bebaiotes*) have also lost their *Sulcia* symbiont entirely, with only one additional recorded case of *Sulcia*-but-not-*Vidania* loss through apparent fungal replacement in *Issus*

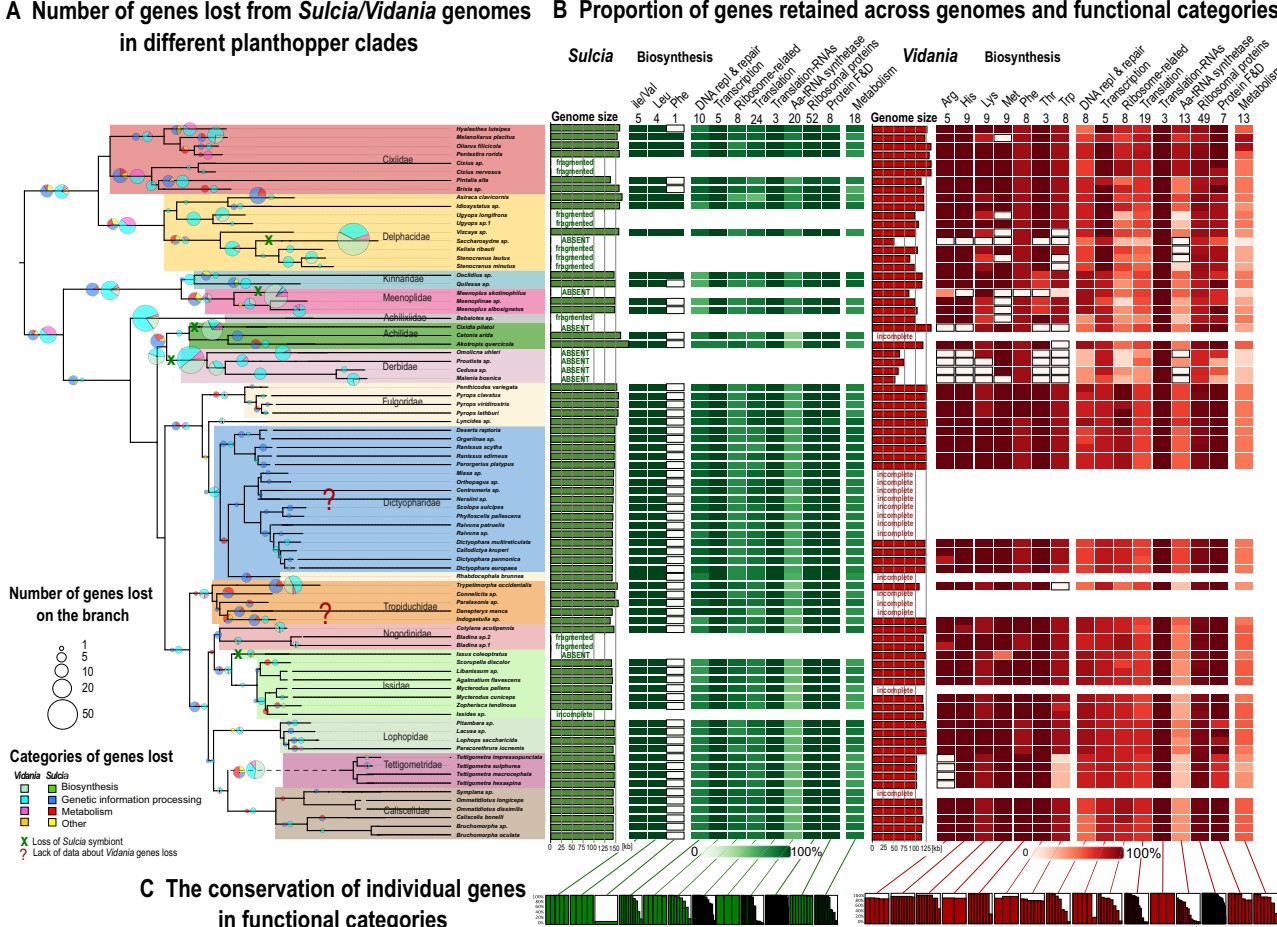

**A** Number of genes lost from *Sulcia/Vidania* genomes in different planthopper clades

**B** Proportion of genes retained across genomes and functional categories

**C** The conservation of individual genes in functional categories

**Fig. 3 | Patterns of gene retention and loss in *Sulcia* and *Vidania* across planthoppers.** The reconstructed patterns of *Sulcia* and *Vidania* gene retention across the planthopper phylogeny indicate gradual gene losses during host-symbiont co-diversification, and much more dramatic changes in some lineages, with substantial variation among symbionts and gene categories. **A** The insect phylogenomic tree, trimmed from Fig. 1 to only include species with a complete genome of at least one symbiont, is used to map the reconstructed numbers of gene losses, indicated by size-scaled pie charts for each symbiont on the branches. Branch for the family Tettigometridae was shortened for the ease of the visualisation. **B** For all complete *Sulcia* and *Vidania* genomes and functional categories, we show the retained proportion of the ancestral gene set. **C** The conservation of individual genes, i.e., the proportion of genomes where the gene is retained, varies among gene categories.

*coleoptratus* (Issidae) lineage. In turn, the branch leading to *Trypetimorpha occidentalis* (Tropiduchidae) and the ancestral branch of Tettigometridae exemplify cases where both *Sulcia* and *Vidania* lost multiple genes (Fig. 3A, B, and Supplementary Figs. 4 and 5).

We also found major differences among gene functional categories in the overall level of conservation (Fig. 3C). In *Sulcia*, ancestral biosynthesis genes are always retained, with one exception: *aspC*, involved in phenylalanine biosynthesis, is found in *Sulcia* from seven species, whereas in most other planthoppers this gene is encoded within the host or *Vidania* genome instead (Supplementary Figs. 4 and 5). In *Vidania*, despite overall conservation, every biosynthetic pathway has been lost from at least one genome, with the Phe pathway most consistently retained. In both symbionts, rRNA genes are always retained, and ribosomal proteins together with genes involved in protein-folding and degradation are highly conserved. In contrast, aminoacyl-tRNA synthetases are the functional category most frequently lost.

**Convergence in the evolution of extremely reduced genomes**

Planthopper symbionts include the smallest bacterial (non-organellar) genomes described to date: that of *Vidania* strain VFSACSP1 from *Saccharosydne* sp. (Delphacidae) has 50,141 bp, and that of VFMALBOS from *Malenia bosnica* (Derbidae) has 52,460 bp (Fig. 4A). VFSACSP1 encodes just 68 protein-coding genes with identified functions, and

VFMALBOS 62. Notably, the extremely reduced state represented by these two genomes has evolved independently in two superfamilies separated by ~263 my of evolution[14].

Despite their independent evolution from an ancestor which must have had a genome approximately three times their size, these smallest genomes exhibit striking similarity in organization and gene contents (Fig. 4A, B, and Supplementary Figs. 2, 5 and 6). The set of genes involved in essential cellular processes comprises rRNA genes, a set of 35 or 38 ribosomal proteins, 8 or 6 DNA and RNA polymerase subunits, 15 other genes involved in genetic information processing, and one or two genes related to metabolism in VFSACPS1 and VFMALBOS, respectively. Both these strains encode the entire ancestral set involved in phenylalanine biosynthesis, while pathways for the biosynthesis of six other essential amino acids have become entirely lost (Fig. 4A, and Supplementary Figs. 5 and 6). Interestingly, neither genome retains any aminoacyl-tRNA synthetases. As in all other *Vidania* genomes, in VFMALBOS, but not in VFSACPS1, tRNAscan-SE identified some tRNA genes, although extreme rates of sequence evolution likely complicated their detection (Supplementary Data 4).

In contrast, the smallest genome from the third recognized planthopper superfamily, of *Vidania* VFMEESKO from a cave-dwelling *Meenoplus skotinophilus* (Meenoplidae), is substantially larger at 84,795 bp and retains an expanded set of genetic information processing and metabolism genes (Fig. 4A). It preserves the complete

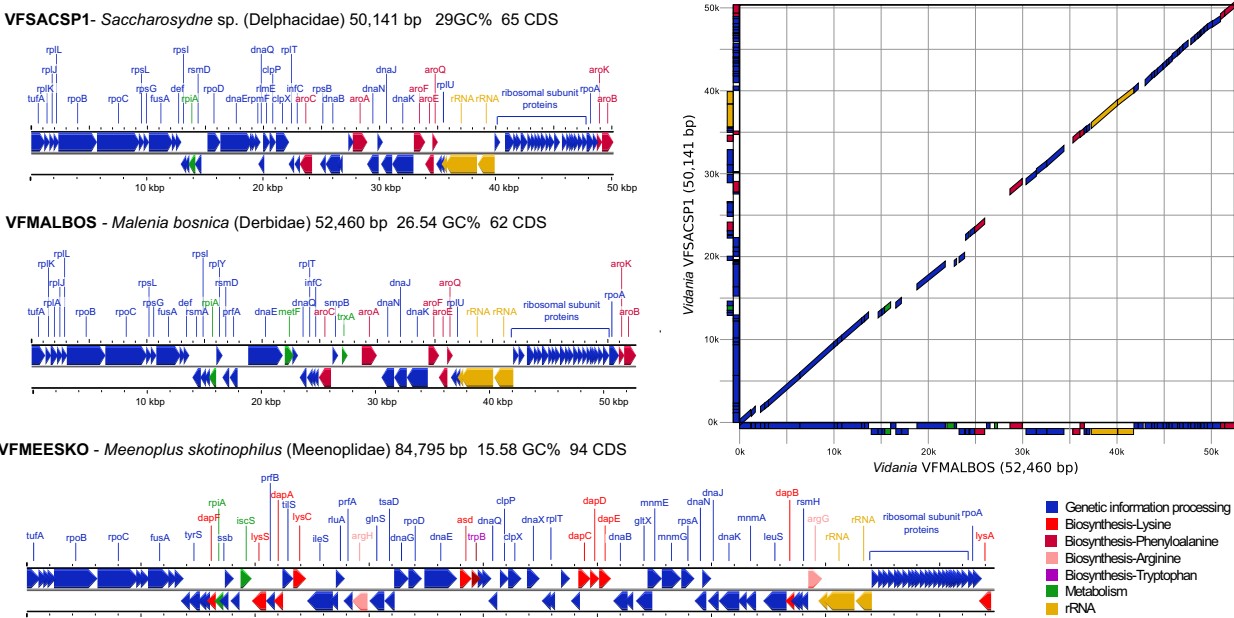

**A** Organization of the tiniest *Vidania* genomes

**B** Congruent gene set in the smallest *Vidania* genomes

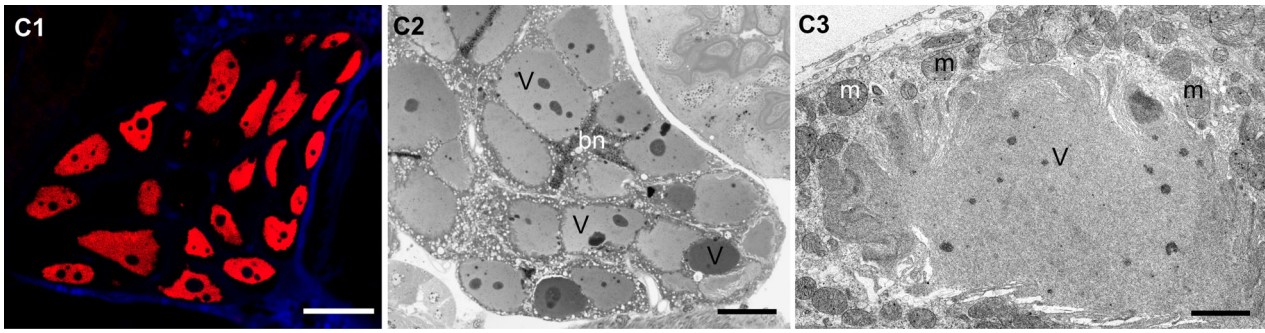

**C** Tissue localization and ultrastructure of *Vidania* cells in *Malenia bosnica*

**Fig. 4 | Genomic and cellular features of highly reduced *Vidania* symbionts.**
**A** The representation of the smallest *Vidania* genomes from three planthopper superfamilies showcases their strongly reduced gene sets. **B** The alignment of the two smallest *Vidania* genomes, which evolved independently from an ancestor approximately three times their size, shows striking convergence in their organization and content. Due to the low sequence similarity among genomes, the alignment plot is based on the positions of annotated genes. **C** The morphology and ultrastructure of bacteriome tissue inhabited by *Vidania* with the smallest genomes differ from previously studied planthoppers by particularly high mitochondrial density - as exemplified by *Malenia bosnica* bacteriome and its tiny-genome *Vidania* VFMALBOS using fluorescence (C1), light (C2), and transmission electron microscopy (C3). The microphotographs were selected as representative from among multiple images obtained for ten adult individuals of *M. bosnica*. bn - bacteriocyte nucleus, m - mitochondrion, V - *Vidania* cell; C1, C2 – 10 μm, C3 – 1 μm.

pathway for lysine biosynthesis and three genes from the arginine and tryptophan pathways, but interestingly, this is the only *Vidania* in our collection that has lost the phenylalanine biosynthesis pathway (Figs. 3B, 4A, and Supplementary Figs. 5 and 6).

Notably, all planthoppers hosting these highly reduced *Vidania* strains have also lost *Sulcia*: they represent the few clades in our dataset where *Sulcia* was lost while *Vidania* remained (Figs. 3 and 5).

The organization and ultrastructure of *Vidania*-hosting bacteriome tissue of *M. bosnica* - the tiny-genome species with several specimens available for microscopy - differed from that of planthoppers studied before[17,19,21] by exhibiting an unusually high density of mitochondria (Fig. 4C). Additionally, in some specimens we observed signs of cellular degeneration, including vacuolization of the bacteriocyte cytoplasm and higher bacterial cytoplasm density (Fig. 4C). These features were also observed in the *Vidania* bacteriomes of other adult planthoppers where this symbiont has reduced gene set and *Sulcia* is absent, including the derbids *Proutista* sp. and *Panmendanga* sp., as well as the achilid *C. pilatoi* (Supplementary Fig. 7). In the bacteriome,

we did not detect any other adjacent bacteria that could serve as an obvious source of proteins targeted to *Vidania* to replace the functions it lost[30].

## Nutritional functions explain the evolution of symbiont genomes and multi-partite symbioses

In hemipteran nutritional symbioses, the set of functions related to nutrient biosynthesis is generally conserved at the level of symbiosis, even when microbial partners change. In previously characterized planthopper symbioses, *Sulcia* and *Vidania* were thought to fulfill their hosts' nutritional needs by jointly encoding most genes for the biosynthesis of all ten essential amino acids, with genes encoded on the host genome thought to complete the pathways (Fig. 5). However, we also find amino acid biosynthesis genes in other bacteria present in these metagenomes, including *Sodalis*, *Arsenophonus*, *Symbiopectobacterium*, *Wolbachia*, or *Tisiphia* (heritable endosymbionts previously detected in planthopper tissues[19]), or *Serratia*, *Klebsiella*, *Sphingomonas*, or *Pseudomonas* (likely less specialized gut associates).

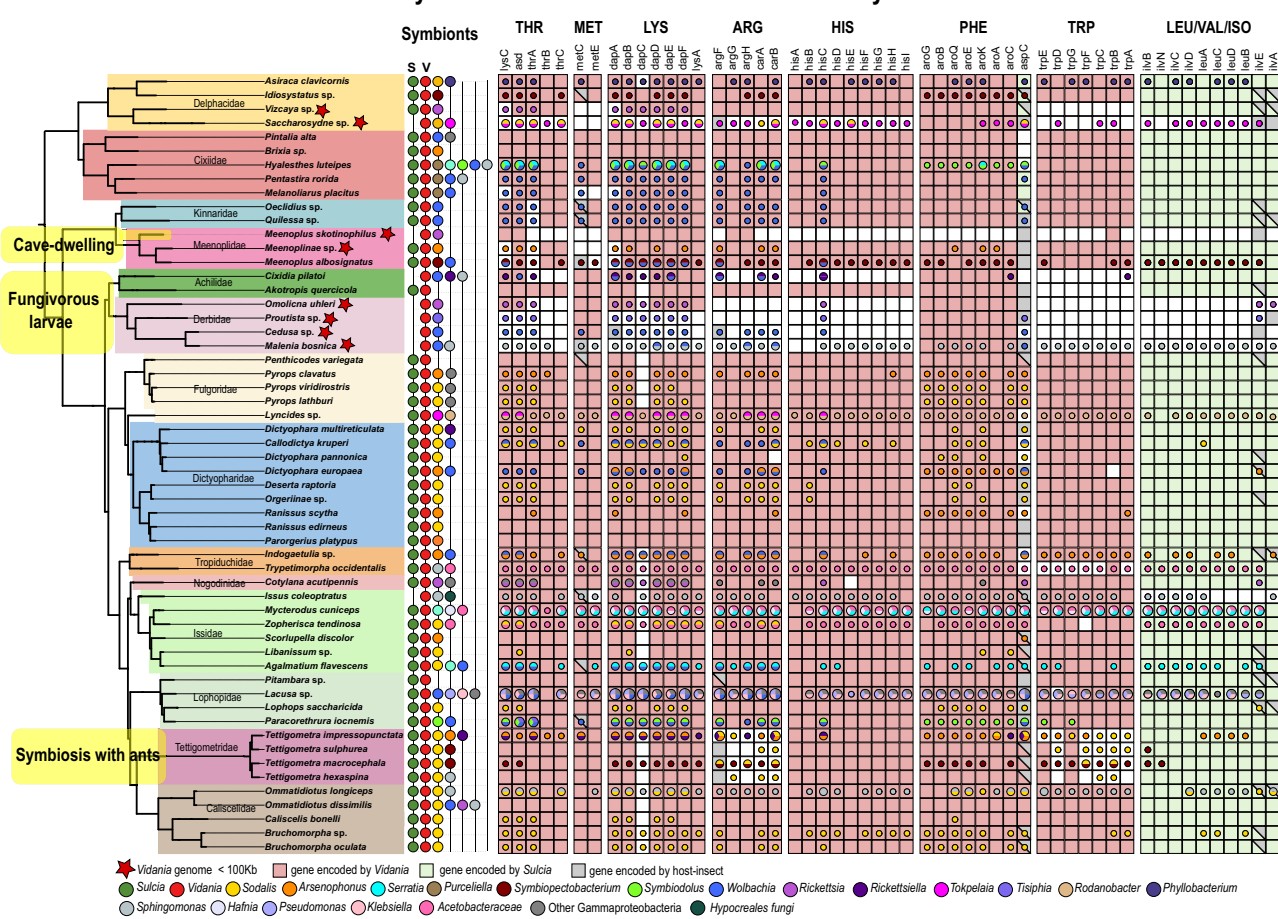

**Fig. 5 | Distribution of amino acid biosynthesis genes across planthopper symbionts.** The distributions of amino acid biosynthesis genes across bacterial genomes in different planthopper species suggest compensation for the loss of pathways from *Vidania* genomes or the loss of *Sulcia*. Host phylogeny is redrawn from Fig. 1, but trimmed to only include species where *Vidania* and *Sulcia* (if present) genomes are complete. Bacterial presence and genus-level classification are based on the combined reconstruction of 16S rRNA and protein-coding genes; *Sulcia* and *Vidania* are represented by colored dots in the columns labelled 'S' and 'V', and other genera, by colors only. Light green, light pink, and grey backgrounds in the main plot panel indicate genes present in *Sulcia*, *Vidania*, and the host genomes, respectively. Colored circles represent genes matched to other bacteria, with the circle size proportional to the number of genomes in which the gene was found.

The genes they encode often overlap with those present in *Sulcia* or *Vidania*, and in many species that had lost *Sulcia*, or some biosynthesis pathways from *Vidania* genomes, they often at least partially fill the resulting gaps (Fig. 5, and Supplementary Data 5). This is most evident in the two planthopper species with the tiniest *Vidania* genomes - *Saccharosydne sp.* and *M. bosnica* - where *Tokpelaia* and *Sphingomonas*, respectively, encode largely complete sets of amino acid biosynthesis genes[22]. However, in assemblies for most other derbids, *Issus coleoptratus*, and others, we did not find alternative copies for a large share of genes lost from ancient genomes. For example, for *M. skotinophilus*, none of the genes from the nine amino acid biosynthesis pathways lost were present in the 695 Mb assembly for its dissected abdomen. We note, however, that functions encoded by microbes residing outside of the tissues used for metagenomics, or sequenced to insufficient coverage, could have been missed. Likewise, the reliable identification of biosynthesis genes encoded within the host genomes requires their more complete assembly.

The biology of most of the studied species is barely known, but mapping significant biological transitions onto the phylogeny suggests the role of host ecology in structuring nutritional demands. Specifically, Derbidae and Achilidae feed on fungal hyphae during their immature stages[31], species in the genus *Tettigometra* engage in obligate trophobiotic associations with ants[32], and *Meenoplus*

*skotinophilus* has adapted to life in caves[33] (Fig. 5). The resulting alterations in nutrient availability and nutritional demands in these clades are likely to explain changes in biosynthetic gene sets.

## Discussion

The symbioses between planthoppers and their ancestral heritable endosymbionts *Sulcia* and *Vidania* can be remarkably stable, as shown by the similarity in symbiont genome organization and contents across the planthopper phylogeny and the previously demonstrated conserved organization of the symbiont-containing tissue[19]. These ancient symbionts have been gradually losing metabolism and genetic information-processing genes during ~263 my of co-diversification with hosts. However, the combination of contributions from other bacteria and host ecological shifts can set the stage for a much more dramatic reduction of *Sulcia* and *Vidania* genomes and functions. In several planthopper lineages, these changes resulted in genomes much smaller than any previously characterized.

The mechanisms driving genomic reduction in insect nutritional intracellular endosymbionts include strong mutation and deletional pressure, the lack of recombination, the absence of opportunities for gene acquisition, and strong genetic drift associated with symbiont transmission bottlenecks[3,4,6]. These processes lead to the gradual loss of non-essential genes, and - according to population-genetic theory,

including Muller's ratchet - the fixation of mildly deleterious but non-lethal changes[8,34,35]. At the same time, the long-term persistence of these associations suggests that host-level selection is sufficiently strong to prevent the symbionts' complete mutational meltdown and halt gene loss, including the retention of symbiont critical cellular functions and the biosynthetic capacity essential to the host. When further gene losses in these highly reduced genomes do occur, they are more likely to reflect changes in host biology or ecological context than purely stochastic accumulation of deleterious mutations.

Nevertheless, additional genes could be lost in three situations: (i) if the host associates with further microorganisms that provide the same function, (ii) host ecological changes render the function unnecessary, or (iii) the function is horizontally acquired by the host[10,36,37]. All three are known evolutionary triggers, and our data suggest that at least the first two may drive symbiont genome degradation in planthoppers. Heritable endosymbionts co-infecting the same hemipteran species are often complementary in their biosynthetic capacity, including cases where genes lost by ancient symbionts are encoded by newer arrivals[10,21,23]. Host nutritional shifts have been convincingly linked to microbiota changes - typically, obligate symbiont gains[37–39]. Our observations on ancient symbiont losses in Achilidae-Derbidae planthoppers parallel similar findings from Typhlocybinae leafhoppers that switched diets from plant sap to parenchymal cell cytoplasm[40].

Ancient heritable endosymbionts and organelles are also known to rely heavily for basic cellular processes on proteins encoded within the host nuclear genome and targeted to organellar or symbiotic membranes or the cytoplasm[6,28,37,41–43]. For example, in *Vidania* genomes, of >30 genes involved in DNA replication in *E. coli*[44] only between eight and two remain. The loss of canonical initiator and helicase components such as *dnaA* and *dnaC*, and key components of constitutive stable DNA replication (cSDR)[45] or rolling-circle replication (RCR)[46,47], indicates that genome replication is likely initiated or regulated by host-encoded factors, while elongation proceeds through the few polymerase subunits that *Vidania* retained (incl. *dnaE, dnaQ*). Similarly, the frequent loss of aminoacyl-tRNA synthetases, key components of the translation process, combined with the retention of tRNAs, suggests that they are likely provided by the host[48]. This functional integration between host and symbiont genetic information processing machineries would parallel the situation in mitochondria, where all core replication and translation proteins are encoded by the host and imported into the organelle[49].

It is hard to envision further losses of highly conserved symbiont metabolism or genetic information processing genes without host involvement, whether re-targeting products of genes previously encoded by the host, including those used to support mitochondria, or alternatively, the acquisition of new functions through gene duplication or horizontal gene transfer[43,50]. Unfortunately, we cannot unequivocally link these alternative mechanisms to specific genomic changes using the current short-read metagenomic data. For clades of interest, we would need dense sampling to comprehend the timing and order of host and symbiont genomic changes and the stability of other microbial associations, advanced tools such as spatial transcriptomics to understand cellular mechanisms, and improved understanding of species' ecology.

To remain viable, a nutritional symbiont must provide biosynthetic functions that make it indispensable for the multi-partner symbiosis, and encode a set of genetic information processing and metabolism genes needed for its basic cellular functions that cannot be provided by the host[4,6]. We showed that in planthopper symbionts, nutritional functions encoded within a genome can be reduced to a single biosynthesis pathway, and the cellular function-related gene set to ca. 55 genes. Non-organellar genomes with more reduced coding capacity are only known from unique symbiotic complexes in cicadas, where genetically distinct cellular lineages

derived from the ancestral *Hodgkinia* symbiont jointly encode the ancestral gene set and appear to exchange functions[5,13].

The striking similarity among independently evolved tiniest *Vidania* genomes suggests that they may have reached the limits of reductive evolution in this specific system. However, judging from the parallels with mitochondria, symbiont genomes could potentially become much smaller. The most reduced known mitogenomes encode two rRNAs and two function-related proteins[29], suggesting that with adequate host support, a theoretical gene content limit of three (rRNAs plus one function-related) may be enough to ensure the genome's long-term persistence[51]. Future genomic surveys across non-model insects may reveal symbionts closer to that limit.

Substantial gene losses can substantially change the symbiont's role in insect biology. The biosynthetic capacity of the most reduced *Vidania* strains is limited to providing a single amino acid, phenylalanine. This precursor of tyrosine, needed for the formation, melanization, and sclerotization of a cuticle - insects' first line of defense against natural enemies and abiotic challenges such as desiccation[52,53]- has been identified as the key symbiotic contribution in beetles and hymenopterans[54–56]. We suspect that the ability to produce lysine, another important cuticle component, by *Vidania* from cave-dwelling *Meenoplus skotinophilus* may reflect host adaptation to an environment where a hardened, melanized cuticle is not necessary. Then, in these planthopper lineages that have also all lost *Sulcia*, the role of ancient nutritional symbiosis may have been reduced to supporting cuticle formation, whereas host insects' other nutritional needs are fulfilled through altered diets or contributions from other microbes.

Planthoppers' reduced reliance on the ancient symbionts is likely to relax selective pressure on their genomic integrity and efficiency, driving ratchet-like accumulation of negative changes and functional losses. This will push the symbiosis deeper into what was described as "the evolutionary rabbit hole" - the state where the host is tied in a mutually obligate relationship with a degenerating partner[8]. Our results expand the range of potential long-term outcomes of such a situation. First, the symbiont undergoing reduction may reach a new 'stable equilibrium' at an even more reduced state, with some functions potentially shifted to other microbes or the host. The striking convergence and similarity among the two tiniest planthopper symbiont genomes suggest that such an equilibrium may exist for *Vidania*, and animal mitochondria demonstrate the potential of long-term stability of extremely reduced genomes[28]. Second, the symbiosis may not be able to escape the degenerative spiral, losing functionality and efficiency, and perhaps ultimately leading to the extinction of the host lineage. The extremely fragmented and very rapidly evolving *Hodgkinia* symbiotic complexes of *Magicicada* periodical cicadas[5], with their apparent high maintenance costs[57], may exemplify this situation. Third, further host ecological changes may render the ancient obligate symbiosis unnecessary anymore: this is where some Typhlocybinae leafhoppers may have arrived[40]. Finally, the replacement of degenerate ancient symbionts by newly arriving and more versatile and efficient bacteria or fungi[18–20] remains a possibility that may enable host organisms to take a new evolutionary path. On the other hand, such replacement is likely to initiate a new cycle of symbiosis degeneration[9], as the "evolutionary rabbit hole" appears to be very difficult to escape from permanently.

## Methods

### Insects

Individuals of 149 specimens representing 19 planthopper families were collected from their natural habitats around the globe between 2004 and 2021 (Supplementary Data 1). Sampling was conducted in accordance with all applicable international and local permitting requirements. This research on non-protected, non-regulated invertebrate species complied with all relevant ethical regulations and institutional guidelines, and required no specific approvals. After

collection, insects were identified based on morphological features and preserved whole in ethanol and stored at −20 °C, or in some cases, partially dissected and fixed in a 2.5% glutaraldehyde solution and stored at 4 °C until further analyses.

## Metagenomic library preparation and sequencing

DNA from separated abdomens or dissected bacteriomes (symbiont-containing tissue) was extracted using the commercially available kits (Supplementary Data 1) and used for metagenomic library preparation using the NEBNext Ultra II DNA Library Prep kit for Illumina (New England BioLabs) or Novogene NGS DNA Library Prep Set, with the target insert length of 350 bp. Due to the known issue of index swapping that occurs during cluster formation and sequencing on Illumina platforms[58,59] and which can lead to cross-contamination among samples in multiplexed lanes, as a means of reducing such cross-talk we used dual-unique indexes for all but six earliest-processed samples. The library pools were sequenced in several batches on Illumina platforms: HiSeq 2500 in a Rapid Run mode (2 × 250 bp), HiSeq X or NovaSeq 6000 S4 (2 × 150 bp reads) (Supplementary Data 1).

## Metagenome assemblies and characterization

Metagenomic raw reads were quality-filtered and adapter-trimmed using Trim Galore v0.6.4 (settings: −length 80 -q 30; https://github.com/FelixKrueger/TrimGalore). High-quality filtered reads were checked using FastQC v0.11.9 (https://github.com/s-andrews/FastQC). Contigs were assembled using Megahit v1.2.9 (maximum k-mer size = 255, min contig size = 1000)[60]. Contigs were validated, and their coverage was estimated by read mapping using Bowtie 2[61]. For the six libraries that were not double-uniquely indexed, we filtered the resulting assemblies for cross-contamination by discarding all contigs that had more than 10X (and typically >200X) greater coverage based on strictly mapped reads from another library with an overlapping index than based on reads from the same library.

Symbiont contigs were taxonomically identified through BLASTN and TBLASTX searches against the curated collection of *Sulcia* and *Vidania* genomes and the NCBI nt database. The circularity and contiguity of *Sulcia* and *Vidania* contigs were confirmed by read mapping and visualization using Tablet v. 1.20.12.24[62], and based on the presence of overlapping ends. The circular *Sulcia* and *Vidania* genomes were rearranged so that they had the same orientation and start position (*Sulcia* - gene *lipB*; *Vidania* - *tufA*). Genomes of symbionts other than *Sulcia* and *Vidania* were represented by multiple contigs and were or will be presented in separate manuscripts[22].

## *Sulcia* and *Vidania* genome annotation

Because of the rapid evolution rates of *Sulcia* and especially *Vidania* genomes and the extreme genetic distance from reference sequences, standard tools such as Prokka or Interproscan struggle to detect and consistently label genes, leaving annotation gaps and unannotated or hypothetical proteins. Therefore, *Sulcia* and *Vidania* genomes were annotated using a custom Python script modified from our previous publication[13]. The script extracts all Open Reading Frames (ORFs) and their amino acid sequences from each genome. It then searches these ORFs recursively using HMMER v3.3.1[63] against custom databases containing manually curated sets of protein-coding, rRNA, and non-coding RNA (ncRNA) gene alignments from previously characterized *Sulcia* or *Vidania* lineages. These references were updated after every round of annotation. rRNA and ncRNA genes were searched with nhmmer (HMMER v3.3.1)[63], and tRNAs were identified with tRNAscan-SE v2.0.7[64]. Based on the relative length compared to the reference genes, protein-coding genes were classified as putatively functional (≥60% of the reference alignment length) or truncated - likely pseudogenes (<60%). These cutoffs were based on our previous work[12,13,17], and we confirmed that key functional domains were preserved in several cases above the cutoff. However, as explained in the section

"Annotation challenges and unusual observations in fast-evolving symbiont genomes" (Supplementary Note 1), the situation was not always clear-cut, and the decisions were made on a case-by-case basis.

Genomes for Fig. 4 were visualized using Proksee[65]. Comparative synteny plots were generated using a custom Python and Processing workflow modified from previous work[13]. Genome content tables were visualized using Processing v3.5.4 and edited with Inkscape v1.4.2.

## Phylogenomic analyses

As the host phylogenomic framework, we used the recently published genome-based planthopper phylogeny[14]. The phylogenies of *Sulcia* and *Vidania* symbionts were reconstructed based on 1291 and 97 protein-coding genes, respectively (Supplementary Data 6). We used *Sulcia* and the betaproteobacterial symbiont *Nasuia* from two species in the Membracoidea superfamily (a treehopper *Entylia carinata* and a leafhopper *Macrosteles quadrilineatus*) as outgroups for planthopper-associated *Sulcia* and *Vidania*, respectively. The single-copy orthologs were identified using OrthoFinder v2.5.5[66] based on the curated proteome resulting from the annotation pipeline above. Orthologs retained by >75% of samples (i.e., by >48 *Sulcia* lineages and by >53 *Vidania* lineages) were used for phylogenomic analyses. Nucleotide alignments of each ortholog were generated by MAFFT v7.526[67]. We kept only the 1st and 2nd codon positions in the alignments to avoid saturation effects from the 3rd codon position. Alignments were then concatenated and partitioned by genes. Phylogenies were inferred with the Maximum Likelihood (ML) approach in IQ-TREE2 v2.3.6[68]. To decide on the partitioning scheme and the substitution model, we asked IQ-TREE2 to perform extended model selection on each gene with free rate heterogeneity and subsequently merge genes until the model fit does not increase any further (setting: -m MFP + MERGE[69]. The best partitioning scheme and the best-fit models were selected based on the highest Bayesian Information Criterion scores. Bootstrapping was conducted using the approximate likelihood ratio test (SH-aLRT) and ultrafast bootstrap methods with 1000 replicates[70,71]. Phylogenetic trees were visualised using iTOL[72].

## Analysis of symbiont contribution to amino acid biosynthesis pathways

The presence of bacterial amino acid biosynthesis genes in genomes of symbionts other than *Sulcia* and *Vidania* was assessed through HMMER searches within six-frame translated assemblies using HMM profiles from the NCBI Protein Family Models database. We taxonomically classified the identified proteins through BLAST-based comparisons against the nt database and against curated *Sulcia* and *Vidania* genomes. At that stage, we also filtered out genes of the known DNA extraction reagent-derived contaminant in our data, *Cellulosimicrobium*. The resulting hits were filtered based on alignment length and sequence similarity, with a large subset manually verified for gene identity and bacterial taxonomic assignment through blastn searches against the NCBI nt database. The results were visualized using a custom Processing script and edited in Inkscape v1.4.2.

## Microscopic analyses

**Histological and ultrastructural analyses.** The whole insect abdomens or dissected bacteriomes were fixed in 2.5% glutaraldehyde in 0.1 M phosphate buffer (pH 7.2) at 4 °C. The fixed material was then rinsed three times in the same buffer with the addition of sucrose (5.8 g/100 ml) and postfixed in 1% osmium tetroxide for 2 h at room temperature. After post-fixation, samples were dehydrated in a graded series of ethanol (30–100%) and acetone, embedded in epoxy resin Epon 812 (Merck, Darmstadt, Germany), and cut into sections using Reichert-Jung ultracut E microtome. Semithin sections (1 μm thick) were stained in 1% methylene blue in 1% borax and analyzed and subsequently photographed under

a Nikon Eclipse 80i light microscope (LM). Ultrathin sections (90 nm thick) were contrasted with uranyl acetate and lead citrate and examined and photographed at 80 kV using a Jeol JEM 2100 electron transmission microscope (TEM) at the Institute of Zoology and Biomedical Research, Faculty of Biology, Jagiellonian University.

**Fluorescence in situ hybridization.** Fluorescence in situ hybridization was performed using fluorochrome-labelled oligonucleotide probes targeting 16S rRNA of *Sulcia*: Sulc644 5′Cy3-CCMCACATTCCAGYTACTCC3′[73] and *Vidania*: Bet940 5′Cy5-TTAATCCACATCATCCACCG3′[74]. Ethanol-preserved insects were rehydrated and then postfixed in 4% paraformaldehyde for 2 h at room temperature. Next, the material was dehydrated again by incubation in increased concentrations of ethanol (30–100%) and acetone, embedded in Technovit 8100 resin (Kulzer, Wehrheim, Germany), and cut into semithin sections (1 μm thick). The sections were then incubated overnight at room temperature in a hybridization buffer containing the specific sets of probes with a final concentration of 100 nM. After hybridization, the slides were washed in PBS three times, dried, covered with ProLong Gold Antifade Reagent (Life Technologies), and examined using a confocal laser scanning microscope Zeiss Axio Observer LSM 710 at the Institute of Zoology and Biomedical Research, Faculty of Biology, Jagiellonian University.

### Reporting summary
Further information on research design is available in the Nature Portfolio Reporting Summary linked to this article.

## Data availability
Raw sequencing data and genomes generated in this study have been deposited in the NCBI database under Umbrella BioProject PRJNA684615, with individual BioProject and BioSample accessions provided in the Supplementary Data 1. The processed data, including genomes, genomic annotations, and sequence alignments used as references in annotation, are available in the Figshare repository under the link https://figshare.com/s/2184f9cfbaae8a21efff[75]. Source data are provided with this paper as a Source Data file.

## Code availability
Annotation, analysis, and visualization scripts are available through the project GitHub page: https://github.com/AnnaMichalik22/Planthopper-ancient-symbioses---supplementary-materials.git.

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

## Acknowledgements

We thank Chris Dietrich, Charles Bartlett, Hannelore Hoch, and Brian Fisher for providing specimens from their collections for this project, Arkadiy Garber for help with the data underlying Fig. 2, and Filip Husnik for valuable discussions. This study was supported by the National Science Centre (Poland) grants 2018/30/E/NZ8/00880 (to P.Ł.) and 2017/

26/D/NZ8/00799 and 2021/41/B/NZ8/04526 (to A.M.) and Polish National Agency for Academic Exchange grant PPN/PPO/2018/1/00015 (P.Ł.). The open-access publication of this article was funded by the programme "Excellence Initiative – Research University" at the Faculty of Biology of the Jagiellonian University in Kraków, Poland.

## Author contributions

A.M.: Conceptualization (equal), Resources (supporting), Methodology (equal), Investigation (lead), Formal Analysis (lead), Visualization (equal), Validation (equal), Data curation (equal), Funding Acquisition (equal), Project Administration (lead), Writing – Original Draft Preparation (supporting), Writing – Review & Editing (equal); D.F.: Data curation (equal), Methodology (equal), Investigation (supporting), Formal Analysis (supporting); J.D.: Investigation (supporting), Formal Analysis (supporting), Visualization (supporting), Data curation (equal); M.P.-F.: Methodology (supporting), Investigation (supporting); A.S.: Resources (lead), Investigation (supporting); P.Ł.: Conceptualization (equal), Methodology (equal), Software (lead), Investigation (supporting), Formal Analysis (supporting), Visualization (equal), Validation (equal), Data curation (supporting), Funding Acquisition (equal), Project Administration (supporting), Writing – Original Draft Preparation (lead), Writing – Review & Editing (equal).

## Competing interests

The authors declare no competing interests.
