## [Transparent Peer Review file · Nature Communications]

Convergent extreme reductive evolution in ancient planthopper symbioses

Corresponding Author: Dr Anna Michalik

Version 0:

Reviewer comments:

Reviewer #1

(Remarks to the Author)

The manuscript "The tiniest genomes shrink much further: extreme reductive evolution in planthopper symbionts" by Michalik and coworkers presents a comparative analysis of the genomes of an extensive group of *Sulcia* and *Vidania* strains, obligate endosymbionts of planthoppers, starting from material that have been collected in a broad survey over the world during many years. This is the result of an impressive amount of work and represents a very comprehensive study that provides some expectable results but also interesting surprises in the evolutionary path of these bacteria.

The way the experiments have been conducted is very appropriated. It is a pity they have not included current long-read sequencing technologies, which would have helped to solve some of the questions that remain unanswered.

The manuscript is well organized, clear and highly informative. I only have a few suggestions that might help improve it:

Abstract, Lines 28-29. Although I agree that the losses undergone by the bacteria with these tiny genomes further blurs the bacteria-organelle boundary, I think it is a bit exaggerated to say that it places them among mitochondria. Due to the yet considerable differences between present endosymbionts of insects and eukaryotic mitochondria, I rather prefer to consider them "symbionelles" as suggested by Reyes-Prieto and coworkers back in 2014 (doi:10.1111/1462-2920.12220). Nevertheless, I must admit that the loss of all aminoacyl-tRNA synthetases is amazing and also common to mitochondria. Do you have any clue about the possibility of the genes encoding them having been transferred to the nucleus of the insect or they are expected to be provided by other bacterial symbionts? Is there any other endosymbiotic model with this feature?

Figure 1. There is no indication of why the five insects depicted in panel A and the one in panel B have been chosen among all species included in the phylogeny.

Lines 390 and 393: There must be an error in the description of the outgroups. As it is written, it indicates that *Sulcia* ENCA is used as the outgroup for *Sulcia* phylogeny.

Supplementary Information

Just driven by curiosity, regarding the example provided in pages 9-10 about the *rplJ* gene: In the figure it appears that *rplA*, which is upstream of *rplJ* in *Vidania* from Akotropis, does not appear in *Vidania* from Brixia. Could be *rplJ2* the consequence of a gene fusion followed by degeneration?

Regarding the broken ORFs (pages 11-12), have you analyzed if the original functional gene encoded a protein with several domains so that the broken version, no matter if it has retained less than 60% of the known functional ortholog, could still be functional? I recall at least one case in an endosymbiont of aphids regarding the annotation of the gene encoding ligase that was considered a pseudogene due to a big truncation that just removed the regulatory domain.

(Remarks on code availability)

I haven't reviewed the code extensively, but it's fully available via GitHub, including a README page.

Reviewer #2

(Remarks to the Author)

This paper was a pleasure to read – it is extremely clear and succinct, both in terms of the writing and figures. I should point out that I'm a little jaded by the "race to the bottom" (of symbiont genome size) that has transpired over the past decade which has yielded many publications in high profile journals that tend to have little content beyond reporting an incrementally smaller genome size. However, this work encompasses a very thorough survey and answers several important outstanding questions in the field, rationalizing concepts involving symbiont replacement, compartmentalization of functions and idiosyncrasies in mutation patterns and rates of degeneration. I therefore think that it merits publication in a high impact journal and will be of interest to a broad readership. To that end, I'll state again that it is written in such a way that it is very clear and broadly accessible. The authors are also appropriate cautious regarding their interpretations and therefore I believe that the results and conclusions are well supported by the observations and data analyses presented in this paper. The figures and tables fit well with the flow and content of the manuscript and are easy to interpret for a broad audience. The methodology is robust and well documented.

I will state that I was really struck by the very limited inventories of genes encoding DNA replication components that are maintained in these small genomes, particularly in light of the fact that they are proposed to replicate circular genomes. It appears to me that many components of the replisome that are specifically involved in "handling" circular chromosome replication (particularly dealing with initiation, termination and dimer resolution are absent, implying that there is a non-canonical method of replication in play. Free-living bacteria that lack such genes tend to replicate by cSDR, but these organisms also don't seem to be capable of initiating cSDR. I would urge the authors to look into this in detail and aim to make some predictions relating to the mode of replication (or at least to outline what is lacking). There are many papers explaining how circular prokaryotic replication systems work and it should be possible to propose a model for operation. This is a very important and interesting question/opportunity, particularly in the light of current work engineering minimal synthetic cells. It could markedly increase the scope for citation of this work.

Some structural suggestions:

Line 35: change to "are much smaller"

Line 54: change to "determine the processes"

Lines 89-90: syntenic/syntenic (not synthetic)?

Line 254: change to "those characterized previously"

One query:

Line 259: What evidence supports the notion that changes are detrimental but not lethal? I think that's an assumption derived from theory rather than empirical evidence and it should be explained as such. Perhaps also mention that these associations have endured for very long periods of time, suggesting that host-level selection is sufficiently potent to ensure that they are not replaced, which is clearly possible, but (IMHO) more likely catalyzed by a changing functional mandate rather than mutational meltdown, as supported by the comparative functional analyses documented in this paper.

(Remarks on code availability)

Reviewer #3

(Remarks to the Author)

NCOMMS-25-58613

The tiniest genomes shrink much further: extreme reductive evolution in planthopper symbionts

Anna Michalik, Diego C. Franco, Junchen Deng, Monika Prus-Frankowska, Adam Stroiński, Piotr Łukasik

General comment

We found this work to be a valuable contribution to the field of insect-microbe symbiosis and bacterial genome evolution. The authors have compiled a series of comprehensive analyses on symbiont genomics across a vast number of planthopper species (149 species representing 19 families!). The discovery of new ultra-small 50–52 kb *Vidania* symbiont genomes is striking, which revised our predisposition on the reductive symbiont genome evolution. The data presented are robust and the methods appear sound. Overall, the manuscript is well-written and provides a framework for driving future studies. On the other hand, there appear to be a few areas where the paper's messages could be strengthened. We provide several comments for making the manuscript more appealing, although those amendments are not mandatory.

Major comments

1. A critical point is that the central question raised in this paper seems somewhat vague, which is a common challenge with this type of exploratory genomic approach. As is common in genomic studies, the initial hypothesis can sometimes become less defined as a result of a data-driven approach. While this is natural, we suppose that the authors could enhance the paper's flow by providing hypothesis uniquely testable in this taxon, or add a clear rationale for their chosen model system. For instance, there is no explicit explanation in the Introduction as to why the authors selected the infraorder Fulgoromorpha to provide a comprehensive view (Line 46-48). A convincing explanation for why planthoppers are uniquely suited for this study would strengthen the paper's main argument.

2. A potential weakness of this study is that it might be perceived as a finding specific only to planthoppers, thereby limiting its broader appeal. A possible way to address this is to provide a strong rationale for why a comprehensive analysis of this

specific group was needed (this is related to Major comment 1). While the authors' discussion already hints at this (Lines 246-249, 269-272, 300-310), we suggest to clearly state why planthoppers were the ideal model for this research in Introduction. This will help readers understand that the findings are not just about a single insect taxon, but provide valuable insights into the fundamental processes of genomic evolution.

3. It is interesting but elusive how the extremely tiny-genome bacterium like *Vidania* can function as a living cellular entity. We would like to see authors' hypothesis/opinion/perspective on the topic in Discussion in a more explicit manner.

Minor Comments

Lines 75-82 (Fig. 1): Please provide more detailed caption for Fig. 1. In Fig. 1A, the color coding for the circles needs to be explained either within the figure or in the caption. For instance, we can assume yellow represents a third obligate symbiont, gray a facultative-like symbiont, and the golden-brown a yeast-like symbiont, but this is not explicitly stated. Additionally, the caption for the confocal microscopy image (Fig. 1C) should specify what the blue color represents.

Lines 104-105 (Fig. 2A): To ensure consistency, please unify the terminology between the figure and its caption. The caption refers to "the very smallest insect symbionts (pink)" and "the smallest bacteria (blue)," but the figure itself uses "insect symbiotic bacteria" and "other bacteria."

Line 124: The authors state they found no cases of gene gain in any lineage. Please clarify the data that supports this statement. Does this mean that no lineage-specific orphan genes were found?

Lines 169-170: To avoid misinterpretation, please use a more precise expression than a range. For the two tiniest genomes, VFSACSP1 and VFMALBOS, it would be clearer to state the number of ribosomal proteins as "35 or 38, respectively." The same applies to the "3-4 ribosomal DNA and RNA polymerase subunits."

Lines 186-192: The imaging results for the tiniest *Vidania* are highly interesting, but the derived observations are open to interpretation. Please clarify the number of samples on which the statements regarding mitochondrial density and vacuolization are based. Additionally, providing context on the host insect's condition (e.g., larva vs. adult) would be valuable. Ideally, a comparison with a closely related insect hosting a non-tiniest *Vidania* would make these findings more conclusive.

Line 188, 190: Fig. 4C?

Line 203: mitochondrium -> mitochondrion

Line 215: Please remove the comma after "Pseudomonas."

Line 355-356: Please state the specific software used for read mapping. This detail is essential for the reproducibility of your analysis.

Line 386: Please add details for the "Processing" and "Inkscape" software, as their specific versions should be specified for reproducibility.

Comments to SI

Lines SI 197-201: We acknowledge the difficulty of accurately annotating highly divergent genes. However, we are not fully convinced by the conclusion that highly truncated genes are functional simply based on their conservation across lineages. Please provide a clearer rationale or discuss other evidence that supports their functionality.

Lines SI 192, 208, 216, 246: We noticed distracting red lines under scientific names in the figures. Please remove them, as they appear to be from an active spell-check.

(Remarks on code availability)

Reviewer #4

(Remarks to the Author)

(Remarks on code availability)

Version 1:

Reviewer comments:

Reviewer #1

(Remarks to the Author)

This is the second time I've reviewed this work. The authors have responded satisfactorily to all my suggestions and comments on the first version, so I have nothing further to add. Congratulations on an excellent job.

(Remarks on code availability)

Reviewer #2

(Remarks to the Author)

From my perspective, the authors have successfully addressed the questions and concerns raised by reviewers, where possible, and the manuscript is greatly improved. I support the publication of this paper in Nature Communication and believes that it will be of broad value to the symbiosis community and beyond.

(Remarks on code availability)

No comments

Reviewer #3

(Remarks to the Author)

We have read the revised version of the manuscript and the authors' responses to the comments. In conclusion, the authors have properly addressed our previous comments. We are satisfied with the revision and have no further major concerns regarding the work. The manuscript is now considered to have reached a standard suitable for publication in this journal.

Specifically, the Introduction section has been substantially revised and significantly improved, clarifying the context and rationale of the study as we suggested. We also appreciate the addition of the TEM data. The inclusion of nice comparative controls is valuable for robust interpretation, even though these may not be the primary focus of the study. Furthermore, all minor points, including corrections to figure captions, have been thoroughly addressed, and I think that the additional discussion is appropriate and strengthens the manuscript's conclusions.

(Remarks on code availability)

Reviewer #4

(Remarks to the Author)

(Remarks on code availability)

RESPONSE TO REVIEWERS' COMMENTS

Reviewer #1 (Remarks to the Author):

The manuscript “The tiniest genomes shrink much further: extreme reductive evolution in planthopper symbionts” by Michalik and coworkers presents a comparative analysis of the genomes of an extensive group of *Sulcia* and *Vidania* strains, obligate endosymbionts of planthoppers, starting from material that have been collected in a broad survey over the world during many years. This is the result of an impressive amount of work and represents a very comprehensive study that provides some expectable results but also interesting surprises in the evolutionary path of these bacteria.

The way the experiments have been conducted is very appropriated. It is a pity they have not included current long-read sequencing technologies, which would have helped to solve some of the questions that remain unanswered.

The manuscript is well organized, clear and highly informative. I only have a few suggestions that might help improve it:

>>> Our response: We thank the reviewer for this valuable comment. We fully agree that long-read sequencing could further resolve ancient symbiont genome organization and evolution - especially for systems where we could not assemble complete *Sulcia* and *Vidania* genomes. Unfortunately, obtaining sufficient amounts of DNA for long-read sequencing has proven highly challenging for these small insects, and due to sample age and varying storage conditions, the DNA has often been degraded and thus unsuitable for long-read technology. Thankfully, for the large majority of ancient symbiont strains analyzed here, the short-read data were sufficient to yield complete, circularly mapping genomes, allowing robust comparative analyses.

We have added the following explanation to the Results: “We note that some of the *Sulcia* and *Vidania* genomes that we were not able to fully assemble based on short-read data, with insufficient DNA amount and quality for long-read sequencing, also appear to have undergone structural changes.” (lines 127-129 in the manuscript version with changes tracked in the manuscript).

We are currently implementing long-read sequencing and complementary approaches such as single-cell sequencing in ongoing projects that resolve fragmented symbiont genomes in a small number of hemipteran species sampled more easily, including two from this study. However, as these projects have a distinct scope and are still in progress, we plan to present their results in separate publications.

Abstract, Lines 28-29. Although I agree that the losses undergone by the bacteria with these tiny genomes further blurs the bacteria-organelle boundary, I think it is a bit exaggerated to say that it places them among mitochondria. Due to the yet considerable differences between present endosymbionts of insects and eukaryotic mitochondria, I rather prefer to consider them

“symbionelles” as suggested by Reyes-Prieto and coworkers back in 2014 (doi:10.1111/1462-2920.12220).

>>> Our response: Thank you for this thoughtful comment. We agree that the term “symbionelle” captures well the highly reduced but still bacterial nature of these intracellular entities. However, as this term has not been widely accepted, we have updated the final sentence of the Abstract using a different wording. The sentence now reads, “Losing many additional cell-function genes places them very close to organelles of symbiotic origin in the level of host dependence, further blurring the bacteria-organelle boundary.” (lines 28-30).

Nevertheless, I must admit that the loss of all aminoacyl-tRNA synthetases is amazing and also common to mitochondria. Do you have any clue about the possibility of the genes encoding them having been transferred to the nucleus of the insect or they are expected to be provided by other bacterial symbionts? Is there any other endosymbiotic model with this feature?

>>> Our response: Thank you for this excellent point. The loss of all aminoacyl-tRNA synthetases and other genes essential for translation has indeed been observed in several other insect symbioses, including *Sulcia–Nasuia/Hodgkinia* in leafhoppers and cicadas, *Tremblaya–Moranella* in mealybugs, and *Buchnera* in aphids. In these systems, the missing functions are likely complemented by host-encoded genes, often of bacterial origin, that are expressed in symbiont-containing tissues and whose products are imported into the symbiont cells (although other possibilities exist, including import of proteins encoded in other symbiont genomes, or of charged tRNAs into symbiont cells). We anticipate a mechanism where host-derived enzymes likely substitute for missing symbiont functions in planthopper symbioses. However, because the host genomic data available for this study are low-coverage and highly fragmented, we were not able to confidently identify horizontally transferred symbiont-support genes.

We have clarified this point in the Discussion and incorporated aminoacyl-tRNA synthetases into the section addressing host support. The revised text reads: “Similarly, the frequent loss of aminoacyl-tRNA synthetases, key components of the translation process, combined with the retention of tRNAs, suggests that they are likely provided by the host⁴⁸. This functional integration between host and symbiont genetic information processing machineries would parallel the situation in mitochondria, where all core replication and translation proteins are encoded by the host and imported into the organelle⁴⁹ “ (lines 340-345).

Figure 1. There is no indication of why the five insects depicted in panel A and the one in panel B have been chosen among all species included in the phylogeny.

>>> Our response: In Figure 1, which provides a general overview of the diversity of both planthoppers and their symbionts, we aimed to present representatives from various planthopper families that differ in morphology and microbiome composition. The final selection was influenced by the availability of high-quality photographs of the planthoppers.

Akotropis quercicola was chosen for Figure 1B based on the quality and availability of FISH images showing the bacteriomes that host the ancestral planthopper symbionts *Sulcia* and *Vidania*. Our prior work has demonstrated that the symbiont morphology and symbiotic tissue organization highly resemble those of other lineages of *Sulcia* and *Vidania*. We have modified this portion of the legend to read: “*Sulcia* (green) and *Vidania* (red) live within the cytoplasm of dedicated cells (bacteriocytes) that form specialized organs (bacteriomes) within the host body cavity. Here, *Akotropis quercicola* (Achilidae) exemplifies the conserved symbiont morphology and bacteriome organization across taxonomically diverse planthoppers” (lines 103-108).

Lines 390 and 393: There must be an error in the description of the outgroups. As it is written, it indicates that *Sulcia* ENCA is used as the outgroup for *Sulcia* phylogeny.

>>> Our response: There is no error; indeed, *Sulcia* and *Nasuia* from treehopper *Entylia carinata* (ENCA) and the leafhopper *Macrostoteles quadrilineatus* (ALF) were used as outgroups for the phylogenies of planthopper-associated *Sulcia* and *Vidania* strains. However, we have indeed not explained that we used non-planthopper *Sulcia* strains to root the planthopper-*Sulcia* tree, and we suspect this may have led to confusion.

We have rewritten that portion of the Results so that it now reads “We used *Sulcia* and the betaproteobacterial symbiont *Nasuia* from two species in the Membracoidea superfamily (a treehopper *Entylia carinata* and a leafhopper *Macrostoteles quadrilineatus*) as outgroups for planthopper-associated *Sulcia* and *Vidania*, respectively.” (lines 466-469) We also removed the branches corresponding to the outgroups from the trees shown in Supplementary Fig. 1 to improve visibility. We have added this information to the figure legend.

Supplementary Information

Just driven by curiosity, regarding the example provided in pages 9-10 about the *rplJ* gene: In the figure it appears that *rplA*, which is upstream of *rplJ* in *Vidania* from *Akotropis*, does not appear in *Vidania* from *Brixia*. Could be *rplJ2* the consequence of a gene fusion followed by degeneration?

>>> Our response: Thank you for your question. We have identified the variant of the *rplJ* gene that we labeled *rplJ2* in five species from different families. In four of them, the *rplA* gene was also present. The length of the *rplJ2* gene is variable, ranging from 145 to 174 nucleotides. Additionally, the *rplJ2* variant of the *rplJ* gene does not show similarity to the *rplA* gene. However, it should be noted that detecting such homology is very difficult due to the very high sequence divergence in these ancient symbionts and the minimal similarity even among variants of the same gene. To address potential further concerns, we have added another example (*Cixius nervosus*) to the Supplementary Information, where we describe this issue in the context of genome annotation (Supplementary Text, page 10).

Regarding the broken ORFs (pages 11-12), have you analyzed if the original functional gene encoded a protein with several domains so that the broken version, no matter if it has retained less than 60% of the known functional ortholog, could still be functional? I recall at least one

case in an endosymbiont of aphids regarding the annotation of the gene encoding ligase that was considered a pseudogene due to a big truncation that just removed the regulatory domain.

>>> Our response: We thank the reviewer for this valuable suggestion. We agree that domain-level analyses can help identify truncated genes that may retain partial functionality. While we did not conduct a systematic domain-level survey, we have now examined the ribosomal protein rplI, for which two variants differing in length were detected among *Sulcia* genomes. The truncated variant retains the key N-terminal domain, suggesting that the encoded protein remains functional. This analysis has been added to the Supplementary Information (Supplementary Text, pages 11-12).

Our custom annotation pipeline for *Sulcia* and *Vidania* uses length thresholds relative to full-length reference genes to classify loci as functional or pseudogenized. We have clarified this in the Methods, adding an explanation that: “These cutoffs were based on our previous work^{12,13,17}, and we confirmed that key functional domains were preserved in several cases above the cutoff”. (lines 454-455). We note that, unlike in cicada symbionts, in planthoppers, in the vast majority of cases, there was no doubt: the gene is either virtually intact or represented by small fragments unlikely to retain function.

Reviewer #1 (Remarks on code availability):

I haven't reviewed the code extensively, but it's fully available via GitHub, including a README page.

>>>Our response: We appreciate the comment. Indeed, all custom scripts used in this manuscript are fully described and available on GitHub.

Reviewer #2 (Remarks to the Author):

This paper was a pleasure to read – it is extremely clear and succinct, both in terms of the writing and figures. I should point out that I’m a little jaded by the “race to the bottom” (of symbiont genome size) that has transpired over the past decade which has yielded many publications in high profile journals that tend to have little content beyond reporting an incrementally smaller genome size. However, this work encompasses a very thorough survey and answers several important outstanding questions in the field, rationalizing concepts involving symbiont replacement, compartmentalization of functions and idiosyncrasies in mutation patterns and rates of degeneration. I therefore think that it merits publication in a high impact journal and will be of interest to a broad readership. To that end, I’ll state again that it is written in such a way that it is very clear and broadly accessible. The authors are also appropriate cautious regarding their interpretations and therefore I believe that the results and conclusions are well supported by the observations and data analyses presented in this paper. The figures and tables fit well with the flow and content of the manuscript and are easy to interpret for a broad audience. The methodology is robust and well documented.

>>> Our response: We sincerely thank the reviewer for this generous and encouraging assessment of our work. From the outset, our goal was to reconstruct the broader patterns and evolutionary drivers underlying the extreme genome reduction observed in planthopper symbionts through comparisons among independent instances of reduction, rather than simply report very small genomes. We are glad that our approach and presentation have conveyed this perspective clearly. At the same time, we agree that the discovery of independently evolved genomes as small as ~50–52 kb adds additional interest and relevance to the study.

I will state that I was really struck by the very limited inventories of genes encoding DNA replication components that are maintained in these small genomes, particularly in light of the fact that they are proposed to replicate circular genomes. It appears to me that many components of the replisome that are specifically involved in “handling” circular chromosome replication (particularly dealing with initiation, termination and dimer resolution) are absent, implying that there is a non-canonical method of replication in play. Free-living bacteria that lack such genes tend to replicate by cSDR, but these organisms also don’t seem to be capable of initiating cSDR. I would urge the authors to look into this in detail and aim to make some predictions relating to the mode of replication (or at least to outline what is lacking). There are many papers explaining how circular prokaryotic replication systems work and it should be possible to propose a model for operation. This is a very important and interesting question/opportunity, particularly in the light of current work engineering minimal synthetic cells. It could markedly increase the scope for citation of this work.

>>> Our response: We thank the reviewer for this insightful comment. Our analyses indeed revealed a striking reduction of replication-associated genes in *Vidania*. Of the >30 genes typically involved in *E. coli* DNA replication, only eight were detected in any *Vidania* genome, and the smallest ~50 kb genomes retain only *dnaE* and *dnaQ*. Genes essential for canonical replication initiation (*dnaA*, *dnaC*) and for alternative mechanisms such as cSDR or rolling-circle replication (*recA*, *priA*, *priB*, *dnaT*, *rep*) are absent, suggesting that *Vidania* cannot sustain autonomous replication.

Because endosymbionts with highly reduced genomes are known to depend on host-derived proteins for essential functions, *Vidania* replication likely proceeds through a host-assisted process analogous to that of mitochondria. We have expanded the Discussion to reflect this interpretation (lines 335-345). This section now reads: “For example, in *Vidania* genomes, of >30 genes involved in DNA replication in *Escherichia coli*⁴⁴ only between eight and two remain. The loss of canonical initiator and helicase components such as *dnaA* and *dnaC*, and key components of constitutive stable DNA replication (cSDR)⁴⁵ or rolling-circle replication (RCR)^{46,47}, indicates that genome replication is likely initiated or regulated by host-encoded factors, while elongation proceeds through the few polymerase subunits that *Vidania* retained (incl. *dnaE*, *dnaQ*). Similarly, the frequent loss of aminoacyl-tRNA synthetases, key components of the translation process, combined with the retention of tRNAs, suggests that they are likely provided by the host⁴⁸. This functional

integration between host and symbiont genetic information processing machineries would parallel the situation in mitochondria, where all core replication and translation proteins are encoded by the host and imported into the organelle⁴⁹.”

Some structural suggestions:

Line 35: change to “are much smaller”

Line 54: change to “determine the processes”

Lines 89-90: synteny/syntenic (not synthetic)?

Line 254: change to “those characterized previously”

>>> Our response: Thank you, the suggested changes have been made.

One query:

Line 259: What evidence supports the notion that changes are detrimental but not lethal? I think that’s an assumption derived from theory rather than empirical evidence and it should be explained as such. Perhaps also mention that these associations have endured for very long periods of time, suggesting that host-level selection is sufficiently potent to ensure that they are not replaced, which is clearly possible, but (IMHO) more likely catalyzed by a changing functional mandate rather than mutational meltdown, as supported by the comparative functional analyses documented in this paper.

>>> Our response: We agree that the assumption that many accumulated changes are “detrimental but not lethal” derives from theoretical predictions rather than direct experimental evidence. We have now revised this section to clarify this point and to emphasize that long-term persistence of these symbioses likely reflects the stabilizing influence of host-level selection, which can mitigate the consequences of mutational accumulation. The revised and expanded section (lines 311-318) now reads: “These processes lead to the gradual loss of non-essential genes, and—according to population-genetic theory, including Muller’s ratchet—the fixation of mildly deleterious but non-lethal changes. At the same time, the very long-term persistence of these associations suggests that host-level selection is sufficiently strong to prevent the symbionts’ complete mutational meltdown and halt the gene loss, including the retention of the symbionts’ critical cellular functions and the biosynthetic capacity essential to the host. When further gene losses do occur, they are more likely to reflect changes in the host biology or ecological context than purely stochastic accumulation of deleterious mutations. “

Reviewer #3 (Remarks to the Author):

General comment

We found this work to be a valuable contribution to the field of insect-microbe symbiosis and bacterial genome evolution. The authors have compiled a series of comprehensive analyses on symbiont genomics across a vast number of planthopper species (149 species representing 19

families!). The discovery of new ultra-small 50–52 kb *Vidania* symbiont genomes is striking, which revised our predisposition on the reductive symbiont genome evolution. The data presented are robust and the methods appear sound. Overall, the manuscript is well-written and provides a framework for driving future studies. On the other hand, there appear to be a few areas where the paper's messages could be strengthened. We provide several comments for making the manuscript more appealing, although those amendments are not mandatory.

>>> Our response: We thank the Reviewer for the positive and encouraging assessment of our work. We appreciate the recognition of the study's scope, data robustness, and significance for understanding symbiont genome evolution. We have carefully considered all of the Reviewer's suggestions and implemented clarifications and additions throughout the revised manuscript to further strengthen the presentation and accessibility of our results.

Major comments

1. A critical point is that the central question raised in this paper seems somewhat vague, which is a common challenge with this type of exploratory genomic approach. As is common in genomic studies, the initial hypothesis can sometimes become less defined as a result of a data-driven approach. While this is natural, we suppose that the authors could enhance the paper's flow by providing hypothesis uniquely testable in this taxon, or add a clear rationale for their chosen model system. For instance, there is no explicit explanation in the Introduction as to why the authors selected the infraorder Fulgoromorpha to provide a comprehensive view (Line 46-48). A convincing explanation for why planthoppers are uniquely suited for this study would strengthen the paper's main argument.

>>> Our response: Thank you for this valuable comment. We agree that clarifying the rationale for focusing on the infraorder *Fulgoromorpha* strengthens the framing of our study. We have now comprehensively revised and expanded the second part of the Introduction to emphasize that planthoppers represent a uniquely powerful model for studying the limits of symbiont genome reduction and replacement (lines 47-69)

2. A potential weakness of this study is that it might be perceived as a finding specific only to planthoppers, thereby limiting its broader appeal. A possible way to address this is to provide a strong rationale for why a comprehensive analysis of this specific group was needed (this is related to Major comment 1). While the authors' discussion already hints at this (Lines 246-249, 269-272, 300-310), we suggest to clearly state why planthoppers were the ideal model for this research in Introduction. This will help readers understand that the findings are not just about a single insect taxon, but provide valuable insights into the fundamental processes of genomic evolution.

>>> Our response: Thank you for this thoughtful suggestion. As noted above, we have substantially expanded the second part of the Introduction. In addition to clarifying why planthoppers provide an exceptional system for studying the limits of symbiont genome reduction, we now emphasize that insights from this group extend beyond a single insect lineage and inform general principles of genome evolution and host–microbe integration.

3. It is interesting but elusive how the extremely tiny-genome bacterium like *Vidania* can function as a living cellular entity. We would like to see authors' hypothesis/opinion/perspective on the topic in Discussion in a more explicit manner.

>>> Our response: We appreciate this thoughtful comment. We agree that understanding how such extremely reduced symbionts remain functional is a key question. We have clarified this point in the revised Discussion, emphasizing that *Vidania* likely relies extensively on host-derived factors for essential cellular functions, including DNA replication (lines 335-345), and that this integration parallels the host control observed in mitochondria. We believe that this addition now explicitly conveys our perspective on how *Vidania* can persist as a living yet highly host-dependent cellular entity.

Minor Comments

Lines 75-82 (Fig. 1): Please provide more detailed caption for Fig. 1. In Fig. 1A, the color coding for the circles needs to be explained either within the figure or in the caption. For instance, we can assume yellow represents a third obligate symbiont, gray a facultative-like symbiont, and the golden-brown a yeast-like symbiont, but this is not explicitly stated. Additionally, the caption for the confocal microscopy image (Fig. 1C) should specify what the blue color represents.

>>> Our response: Thank you for your comment. The colored circles in the figure have been labeled as *Sulcia*, *Vidania*, Gammaproteobacteria, Alphaproteobacteria, and Hypocreales. However, we agree that an additional note in the legend improves clarity, and we have therefore added a concise explanation of the color coding. We have also specified in the caption that the blue color in panel B represents DAPI staining (lines 98-108).

Lines 104-105 (Fig. 2A): To ensure consistency, please unify the terminology between the figure and its caption. The caption refers to “the very smallest insect symbionts (pink)” and “the smallest bacteria (blue),” but the figure itself uses “insect symbiotic bacteria” and “other bacteria.”

>>> Our response: Thank you for pointing this out. We have revised the terminology in the caption to ensure consistency. The figure and the caption now use the same terms: “Fulgoromorpha symbionts”, “insect-symbiotic bacteria”, “other bacteria”. The legend now reads:

“In terms of genome size and gene content, *Fulgoromorpha* symbiont genomes (red) fall among the very smallest known for insect-symbiotic bacteria (pink), which themselves are among the smallest bacterial genomes overall (other, non-insect-symbiotic bacteria shown in blue). Cellular organelles—mitochondria (orange) and chloroplasts (green)—are shown for reference.” (lines 141-144)

Line 124: The authors state they found no cases of gene gain in any lineage. Please clarify the data that supports this statement. Does this mean that no lineage-specific orphan genes were found?

>>> Our response: We thank the reviewer for this valuable comment. By stating that we found no cases of gene gain, we meant that no lineage-specific genes of apparent external origin were detected—that is, no evidence of horizontal gene transfer based on sequence similarity, GC content, or phylogenetic signal.

In this context, we would consider “orphan genes” as lineage-specific open reading frames (ORFs) lacking detectable homologs in any other known organism and not traceable to ancestral genes within the same symbiont lineage. To evaluate this, we used a custom annotation pipeline combining *hmmer* searches with manually curated reference alignments to detect highly diverged homologs and performed *phmmer* searches of all unannotated ORFs >50 amino acids in any of the genomes against the UniProt database. We included these ORFs as references in subsequent rounds of annotation, allowing us to understand their level of conservation.

Across all genomes, we identified 40 *Sulcia* and 86 *Vidania* ORFs with no significant similarity to known proteins - as explained in Results (lines 161-164). These ORFs are typically short, AT-rich, and often conserved among related species, often occupying syntenic positions corresponding to degraded ancestral loci. These characteristics suggest that they represent ancestral genes, or their remnants, rather than newly evolved, lineage-specific orphan genes.

Lines 169-170: To avoid misinterpretation, please use a more precise expression than a range. For the two tiniest genomes, VFSACSP1 and VFMALBOS, it would be clearer to state the number of ribosomal proteins as “35 or 38, respectively.” The same applies to the “3-4 ribosomal DNA and RNA polymerase subunits.”

>>> Our response: Thank you, the change has been made (lines 212-216)

Lines 186-192: The imaging results for the tiniest *Vidania* are highly interesting, but the derived observations are open to interpretation. Please clarify the number of samples on which the statements regarding mitochondrial density and vacuolization are based. Additionally, providing context on the host insect’s condition (e.g., larva vs. adult) would be valuable. Ideally, a comparison with a closely related insect hosting a non-tiniest *Vidania* would make these findings more conclusive.

>>> Our response: We thank the reviewer for this helpful comment. Using TEM, we analyzed ten adult individuals of *Malenia bosnica* and consistently observed a high mitochondrial density in the cytoplasm of *Vidania* bacteriocytes, together with signs of cell degeneration, including vacuolization of the bacteriocyte cytoplasm. Similar features were observed in

Vidania bacteriocytes of other adult species lacking *Sulcia* and harboring only *Vidania* with reduced gene sets, including other derbids and *Cixidia pilatoi* (Achilidae).

To provide additional context and comparison, we have added Supplementary Fig. 7, which shows *Vidania* bacteriocytes from other planthoppers with highly reduced genomes alongside those from species hosting *Vidania* with more complete genomes. The legend of Figure 4 has been updated to highlight that *M. bosnica* represents more general patterns:

“The morphology and ultrastructure of bacteriome tissue inhabited by *Vidania* with the smallest genomes differ from previously studied planthoppers by particularly high mitochondrial density—as exemplified by *Malenia bosnica* bacteriome and its tiny-genome *Vidania* VFMALBOS using fluorescence (C1), light (C2), and transmission electron microscopy (C3)” (lines 245-249). We also revised the Results to note that: “These features were also observed in the *Vidania* bacteriomes of other adult planthoppers where this symbiont has highly reduced genome and *Sulcia* is absent, including the derbids *Proutista* sp. and *Panmendanga* sp., as well as the achilid *Cixidia pilatoi* (Supplementary Fig. 7)”. (lines 234-236)

Line 188, 190: Fig. 4C?

>>> Our response: Thank you for catching this. Of course, we meant to refer to figure 4C rather than 5C, and have corrected this.

Line 203: mitochondrium -> mitochondrion

Line 215: Please remove the comma after “Pseudomonas.”

Line 355-356: Please state the specific software used for read mapping. This detail is essential for the reproducibility of your analysis.

Line 386: Please add details for the “Processing” and “Inkscape” software, as their specific versions should be specified for reproducibility.

>>> Our response: Thank you for pointing these out. We have incorporated all of your suggestions into the text.

Comments to SI

Lines SI 197-201: We acknowledge the difficulty of accurately annotating highly divergent genes. However, we are not fully convinced by the conclusion that highly truncated genes are functional simply based on their conservation across lineages. Please provide a clearer rationale or discuss other evidence that supports their functionality.

>>> Our response: We thank the reviewer for this helpful comment. As explained above, we have expanded the Supplementary Materials (Supplementary Text, lines 228-242) to clarify the rationale for considering some of the truncated genes as likely functional. Using *rplI* gene as an example, we explain how the *rplI2* gene variant encodes a shortened form of the ribosomal protein RplI that retains the essential N-terminal RNA-binding domain, responsible for ribosome association and stability. The conservation of this truncated variant across nine

planthopper families supports its functional relevance rather than random sequence retention. This clarification has also been noted in the main text (lines 452-458).

Lines SI 192, 208, 26, 246: We noticed distracting red lines under scientific names in the figures. Please remove them, as they appear to be from an active spell-check.

>>> Our response: Thank you for pointing these out. We have removed all those red lines.

Reviewer #4 (Remarks to the Author):

>>> Our response: We thank Reviewer #4 for contributing to the review process and appreciate their participation in the *Nature Communications* Early Career Reviewer initiative.